# Assessing the Measurement Properties of the Test of Gross Motor Development-3 Using the COSMIN Methodology—A Systematic Review

**DOI:** 10.3390/bs15010062

**Published:** 2025-01-13

**Authors:** Yuanye Zhu, Jing Wang, Yaru Ding, Yongdong Qian, Mallikarjuna Korivi, Qian Chen, Weibing Ye

**Affiliations:** 1Zhejiang Sports Science Institute, Hangzhou 310004, China; zhuyuanye@zjnu.edu.cn; 2Institute of Human Movement and Sports Engineering, College of Physical Education and Health Sciences, Zhejiang Normal University, Jinhua 321004, China; tyxyqyd@zjnu.cn (Y.Q.); mallik.k5@gmail.com (M.K.); 3Institute of Culture Creativity, Weifang Vocational College, Weifang 261000, China; 2021020074@sdwfvc.edu.cn; 4China Volleyball Sport College, Beijing Sport University, Beijing 100084, China; 2024240912@bsu.edu.cn

**Keywords:** scale, outcome measurement instrument, fundamental movement skills, reliability, validity, responsiveness

## Abstract

This study aimed to systematically review the measurement properties of the Test of Gross Motor Development-3 (TGMD-3) using the COSMIN methodology. A search of four databases (PubMed, EMBASE, Web of Science, CINAHL) identified 23 relevant studies. The methodological quality of the studies was assessed using the COSMIN risk of bias checklist; the measurement properties of the TGMD-3 were evaluated by the COSMIN quality criteria; and the quality of the evidence was rated using a modified GRADE approach. The findings indicated that the test–retest, inter-rater, and intra-rater reliability, as well as measurement invariance and part content validity (relevance and comprehensibility), were sufficient, supported by high-quality evidence. The bifactor structure was found to be a more appropriate model for the TGMD-3, with structural validity and internal consistency rated as sufficient, though based on moderate-quality evidence. However, hypothesis testing for construct validity produced inconsistent results, also supported by moderate-quality evidence. Responsiveness was rated as inconsistent, based on low-quality evidence. Overall, the TGMD-3 is graded as “B”, meaning it has the potential to be recommended, but further research is needed to fully establish its measurement properties. Future studies should focus on verifying the comprehensiveness of items of the TGMD-3 to optimise its application.

## 1. Introduction

Shortly after birth, infants display instinctive behaviours such as sucking and grasping ([61]). As children’s body coordination improves, they gradually learn to lift their heads and roll over, among other motor behaviours. More generally, motor behaviour development brings about new opportunities for acquiring knowledge about the world, and burgeoning motor skills can instigate cascades of developmental changes in perceptual, cognitive, and social domains ([2]). Motor skills development is commonly categorised into two main areas: gross motor skills and fine motor skills. Gross motor skills encompass movements that require the use of large muscle groups, like sitting unsupported, crawling, walking, and running. On the other hand, fine motor skills pertain to the use of smaller muscles for activities such as grasping objects, manipulating them, or engaging in tasks like drawing ([21]). Fundamental movement skills (FMSs) are defined as the “basic learning movement patterns in preschool children” ([22]). These skills encompass essential abilities that enable children to perform structured movements ([60]; [62]), primarily involving gross motor skills such as running, jumping, sliding, striking, catching, and kicking. Milestones provide a framework for observing and monitoring a child over time ([20]). According to a survey by the WHO, generally speaking, infants achieve the milestone of sitting independently between 3.8 and 9.2 months, standing between 6.9 and 16.9 months, and walking between 8.2 and 17.6 months ([58]). By the age of 2, a child possesses the ability to kick a ball, jump with both feet leaving the ground, and throw a large ball overhand ([20]). The milestones for subsequent ages (after the age of 3) indicate advancements in the duration, frequency, or the successful execution distance of each task ([20]). Proficiency in FMSs is crucial for children’s overall development, with numerous studies demonstrating positive correlations between FMSs and various health-related outcomes, including body composition ([37]), academic achievement ([13]), cognitive function ([36]), and social skills ([15]). Inadequate FMS development can hinder children’s ability to participate in physical activities, preventing them from reaching key developmental milestones ([45]). Over the past few decades, the importance of assessing motor competence has gained significant attention, prompting the development of various reliable and valid instruments to accurately screen for FMS ([11]; [16]; [24]). These tools play a central role in promoting physical literacy, informing targeted interventions, and addressing public health concerns related to childhood obesity and sedentary lifestyles ([25]).

All versions of the Test of Gross Motor Development (TGMD)—including TGMD-1, TGMD-2, and TGMD-3—are process-oriented assessments designed to evaluate fundamental movement skills ([52]). The original TGMD was developed by Ulrich in 1985 in his doctoral dissertation to assess FMSs in physical education settings ([49]). The normative sample consisted of 909 children from throughout the US ([59]). Several studies have examined its psychometric properties. [48] ([48]) demonstrated that the TGMD possesses excellent content validity ([48]), while [50] ([50]) proved that this test is highly sensitive in measuring changes in preschool children’s fundamental motor skill performance ([50]). [18] ([18]) support the bifactor structure of the TGMD by assessing the construct validity of the TGMD ([18]). According to national standards for evaluating educational and psychological tests, norm-referenced tests should be re-standardised approximately every 15 years to account for population changes over time ([49]). This led to the creation of TGMD-2 in 2000, which updated the normative sample, and subsequently, the TGMD-3, which incorporated feedback from users to refine the instrument ([49]).

The measurement properties of an instrument, as outlined by the COnsensus-based Standards for the selection of health Measurement INstruments (COSMIN), are essential for determining its reliability, validity, and responsiveness. According to COSMIN, reliability includes test–retest, inter-rater, and intra-rater reliability, while validity encompasses content, construct (including structural, cross-cultural, and hypothesis testing), and criterion validity ([35]). TGMD-3 is now widely utilised in both research and physical education contexts to assess children’s motor competence ([10]; [17]; [27]; [29]; [30]; [31]; [34]; [39]). Since the introduction of the TGMD-3, numerous studies have examined its measurement properties ([17]; [27]; [29]; [30]; [31]; [34]; [39]; [53]; [56]). While many of these studies confirm the reliability and validity of the TGMD-3, inconsistent findings remain. For instance, some research supports a bifactor structure (the scale having two dimensions) for the TGMD-3. Based on this, the scale was divided into two subscales named the locomotor subscale and the ball skills subscale ([7]; [5]; [14]; [27]; [43]; [55]). Whereas other studies argue that a one-factor structure is also appropriate ([17]; [19]; [31]; [34]; [57]), the TGMD-3 cannot be divided into subscales. Furthermore, while some studies have reported sufficient internal consistency for the TGMD-3’s locomotor subscale ([34]), others have found it inadequate ([53]). These inconsistent reports highlight the need for a comprehensive, systematic review to provide clarity on the measurement properties of the TGMD-3.

The COSMIN methodology is widely recognised as a robust framework for evaluating the measurement properties of health-related instruments, offering a systematic approach to assessing their reliability, validity, and responsiveness ([35]). Its application has proven effective in a variety of domains, including motor competence assessments, as it ensures that instruments are both scientifically rigorous and applicable across different populations. For instance, Hulteen et al. utilised the COSMIN framework in their systematic review of motor assessment tools in children and adolescents, revealing key insights into the psychometric properties of these instruments ([23]). The COSMIN methodology has also been applied to single instruments, as demonstrated by reviews of the Body Image Scale ([33]) and the Peabody Developmental Motor Scales-2 ([63]), which confirmed its versatility in evaluating a wide range of scales. Given the inconsistencies in previous studies regarding the TGMD-3’s measurement properties, applying the COSMIN methodology to systematically review the TGMD-3 offers an opportunity to standardise the evidence, ensuring a more reliable and comprehensive understanding of its psychometric robustness. This review aims to consolidate existing research on the TGMD-3 and synthesise the quality of evidence through the COSMIN methodology, thereby offering valuable guidance for both researchers and practitioners in the fields of physical education and child development. By critically examining the reliability, validity, and overall measurement properties of the TGMD-3, this review seeks to determine the extent to which the instrument can be trusted to accurately assess fundamental movement skills of children.

## 2. Materials and Methods

### 2.1. Literature Search Strategy

A systematic search was conducted using English as the search language across four major electronic databases—PubMed, EMBASE, Web of Science, and CINAHL—targeting studies that evaluated the measurement properties of the Test of Gross Motor Development, Third Edition (TGMD-3) up to September 2024. The search strategy was designed to comprehensively capture all relevant studies using an array of search terms associated with the TGMD-3. These included variations in the instrument’s name: the test of gross motor development OR TGMD OR “the test of gross motor development-3” OR “TGMD-3” OR “the test of gross motor development-third edition” OR “the test of gross motor development-3rd” OR “the Test of Gross Motor Development, Third Edition” OR “Test of Gross Motor Development-Third Edition” OR “Test of Gross Motor Development, 3rd Edition”.

To identify studies examining the psychometric properties of TGMD-3, search terms were combined with keywords representing specific measurement properties, such as reliability OR “internal consistency” OR “measurement error” OR validity OR “content validity” OR “face validity” OR “construct validity” OR “structural validity” OR “hypotheses testing” OR “cross cultural validity” OR “criterion validity” OR responsiveness OR “measurement properties” OR “psychometric properties” OR “measurement property” OR “psychometric property” OR “ divergent validity” OR “concurrent validity” OR “predictive validity”.

The search process adhered to the latest Preferred Reporting Items for Systematic Reviews and Meta-Analyses (PRISMA) guidelines ([38]), ensuring a rigorous, transparent, and reproducible approach. Full-text articles of the studies meeting the inclusion criteria were accessed either via the publisher’s platform or, when necessary, through institutional resources or external collaborations. The registration of the study protocol was completed in PROSPERO accessed on 4 November 2024 (https://www.crd.york.ac.uk/prospero/; CRD42024600 851).

### 2.2. Inclusion and Exclusion Criteria

The studies included in this review had to meet the following criteria: (1) studies focusing on typically developing children aged 3 to 11 years or children with disabilities; (2) studies evaluating the measurement properties of the Test of Gross Motor Development-3 (TGMD-3); and (3) studies assessing at least one measurement properties of the TGMD-3.

Studies were excluded if they (1) utilised the TGMD-3 solely to investigate children’s FMSs without evaluating the scale’s measurement properties; (2) utilised the TGMD-3 solely for evaluating the efficacy of the intervention; (3) were reviews, systematic reviews, or meta-analyses; or (4) provided only an abstract, lacked full-text access, or were not peer-reviewed.

### 2.3. Literature Selection and Data Extraction

Two reviewers, YZ and JW, independently handled the process of selecting the literature and extracting data. Discrepancies between the reviewers were resolved through discussion and, if necessary, consultation with a third reviewer (YQ). For any unresolved disagreements, further input was sought from additional review authors (WY and QC).

All identified references were managed using EndNote software (Version 20.2.1), where duplicate records were automatically excluded. The selection process involved two stages. First, the titles and abstracts were screened to remove irrelevant studies. In the second stage, the full texts of the remaining articles were thoroughly reviewed based on the predefined inclusion and exclusion criteria.

From the eligible studies, the following information was systematically gathered: (1) study attributes such as the primary author’s name, publication year, study cohort, geographical area, sample size, and demographics in terms of age and gender; (2) TGMD-3 measurement characteristics assessed, encompassing internal consistency, content and structural validity, cross-cultural validity, measurement invariance, reliability, measurement error, criterion validity, and responsiveness; (3) detailed data pertaining to each evaluated measurement characteristic.

### 2.4. Assessment of Risk of Bias and Evidence Quality in Included Studies

To assess the methodological quality of the included studies, the COSMIN risk of bias checklist ([35]) was employed. This checklist comprises ten areas, namely, “PROM development, content validity, internal consistency, structural validity, measurement invariance/cross-cultural validity, reliability, criterion validity, hypothesis testing for construct validity, measurement error, and responsiveness”. Based on the specific measurement properties discussed in each study, the relevant sections were chosen. The methodological quality of each item was rated as “very good”, “adequate”, “doubtful”, or “inadequate”, adhering to standardised scoring criteria. The overall methodological quality of each study was decided using the “worst score principle”, where the lowest score within a section determined the study’s overall rating. Two reviewers, YZ and JW, independently conducted the bias risk assessment for all articles. Disagreements between the reviewers were resolved through discussion and, if required, by consulting a third reviewer, YQ.

To synthesise the quality of evidence, a modified version of the Grading of Recommendations Assessment, Development and Evaluation (GRADE) method ([35]) was employed. This adaptation is tailored to the COSMIN framework, refining the original GRADE system. Evidence levels were categorised as “high”, “moderate”, “low”, or “very low”. All included studies initially received a “high” level of evidence, with potential downgrades applied based on specific study characteristics. Unlike the conventional GRADE approach, this modified version omits the “publication bias” consideration. The quality of evidence may be reduced due to factors such as risk of bias, inconsistency, indirectness, and imprecision.

### 2.5. Overall Rating of the Measurement Properties

The overall rating for each measurement property of the TGMD-3 was determined following the COSMIN methodology for assessing the content validity of the PROM user manual ([47]) and the COSMIN methodology for systematic reviews of the PROM user manual (COSMIN manual) ([35]).

The evaluation encompassed various measurement properties such as content validity, internal consistency, structural validity, criterion validity, cross-cultural validity/measurement invariance, reliability, measurement error, hypothesis testing for construct validity, and responsiveness, as detailed in Appendix A. Each of the reported items was assessed and categorised as “sufficient (+)”, “insufficient (−)”, or “indeterminate (?)” based on the criteria outlined in Appendix A. The overall rating for each measurement property was then determined as “sufficient (+)”, “insufficient (−)”, “inconsistent (±)”, or “indeterminate (?)”, reflecting the comprehensiveness and rigour of the evaluation process.

In cases where results were inconsistent, further analysis was undertaken to identify potential reasons for these discrepancies. For hypothesis testing of construct validity, the research team pre-defined hypotheses. Specifically, construct convergent or concurrent validity was deemed sufficient when the correlation coefficient between TGMD-3 and a comparator instrument measuring a similar construct was ≥0.50. Construct validity was rated as “sufficient (+)” if 75% or more of the results aligned with the hypotheses, “insufficient (−)” if 75% or more did not, or “indeterminate (?)” if no hypotheses were established.

## 3. Results

### 3.1. Literature Search Results

A total of 1054 articles were retrieved from the database search, including 198 from CINAHL, 57 from PubMed, 737 from Web of Science, and 62 from EMBASE. The search was conducted up to 26 September 2024, with no restriction on the earliest date of publication.

Following the import of all identified articles into EndNote, 313 duplicates were removed. The titles and abstracts of the remaining 741 articles were then screened, resulting in the exclusion of 707 articles due to irrelevance. This initial screening left 34 articles for further consideration. Twelve articles were excluded due to the unavailability of their full texts (being conference abstracts only), and subsequently, twenty-two articles underwent eligibility assessment. Upon further evaluation, five articles were excluded for the following reasons: two studies used the TGMD-3 to assess other scales ([1]; [12]), one study did not evaluate the measurement properties of the TGMD-3 ([6]), and two studies focused on assessing other versions of the TGMD-3 ([14]; [52]). Subsequently, we added six articles by screening the references in the included studies. Ultimately, 23 articles met the inclusion criteria and were included in this systematic review. The figure below (Figure 1) illustrates the comprehensive selection process, outlining the number of articles at each stage of the process.

### 3.2. Characteristics of the Included Studies

Table 1 summarises the key features of the 23 studies included in this review. These studies span various continents, with a significant proportion originating from Europe ([9]; [14]; [17]; [27]; [28]; [30]; [32]; [55]). North American studies also featured prominently ([7]; [5]; [19]; [26]; [29]; [39]; [44]; [57]), alongside contributions from Asia ([34]; [42]; [43]) and South America ([30]; [51]; [53]).

Breaking the included studies down by country, seven studies were conducted in the United States ([7]; [5]; [19]; [26]; [39]; [44]; [57]); three in Spain ([9]; [17]); two in Brazil ([51], [53]); two in Italy ([27]; [28]); and two in Iran ([27]; [28]). Additional studies originated from Australia ([3]), Ireland ([14]), Canada ([29]), Peru ([30]), Slovenia ([31]), Bosnia and Herzegovina ([32]), Indonesia ([42]), and Germany ([55]).

### 3.3. Synthesis of Evidence for the Measurement Properties of TGMD-3

The synthesis of evidence for the measurement properties of the TGMD-3, along with the corresponding quality of evidence for each property, is summarised in Table 2. These findings offer a comprehensive evaluation of the psychometric properties of the instrument, allowing for a more informed understanding of its reliability and validity. Detailed data on the quality of evidence for each measurement property are presented in the Appendix A, providing further insight into the robustness of the results.

#### 3.3.1. Content Validity

Out of the 23 included studies, four specifically assessed the content validity of the TGMD-3 ([30]; [31]; [34]; [53]). These studies systematically evaluated the content validity by consulting experts in the field to determine the relevance and comprehensibility of the TGMD-3. The overall rating for the relevance and comprehensibility of the scale was deemed sufficient, with a moderate quality of evidence. However, since none of the studies addressed the comprehensiveness of items of the TGMD-3, an overall rating for this aspect could not be determined, nor was it possible to provide a qualitative synthesis of content validity (Table 2).

#### 3.3.2. Structure Validity

Thirteen of the twenty-three studies evaluated the structural validity of the TGMD-3 using classical test theory (CTT) ([7]; [5]; [17]; [19]; [27]; [29]; [30]; [31]; [32]; [34]; [43]; [53]; [57]). Thirteen studies explored the bifactor structure of the TGMD-3, with five studies also assessing a one-factor structure. The bifactor model was rated as sufficient, with 79% of studies supporting this structure, and the quality of evidence was considered moderate. In contrast, the one-factor model exhibited inconsistent results, leading to a moderate quality of evidence (Table 2).

#### 3.3.3. Internal Consistency

As summarised in Table 2, thirteen studies assessed the internal consistency of the TGMD-3 ([3]; [7]; [17]; [19]; [28]; [30]; [31]; [32]; [34]; [42]; [53]; [55]; [57]). Cronbach’s alpha values for the Locomotor subscale ranged from 0.63 to 0.92, with 92% of results exceeding the threshold of 0.7, indicating sufficient internal consistency. Similarly, the Ball Skills subscale demonstrated alpha values ranging from 0.60 to 0.95, with 92% of results surpassing 0.7, also rated as sufficient. The Total TGMD-3 scale exhibited alpha values between 0.74 and 0.96, confirming sufficient internal consistency. The overall rating for internal consistency was deemed sufficient, although the quality of evidence was moderate due to some inconsistencies across the studies.

#### 3.3.4. Reliability

Thirteen studies examined the reliability of the TGMD-3 ([3]; [7]; [9]; [17]; [26]; [27]; [30]; [31]; [34]; [53]; [55]; [57]). Following the COSMIN guidelines ([35]), these studies assessed test–retest reliability, inter-rater reliability, and intra-rater reliability.

Seven studies explored the test–retest reliability of the TGMD-3 ([27]; [30]; [31]; [34]; [53]; [55]; [57]). Intraclass correlation coefficients (ICCs) ([3]; [27]; [30]; [31]; [55]; [57]) and Pearson correlation coefficients (r) ([34]; [53]) were the primary metrics used to assess this property. The ICCs ranged from 0.81 to 0.996 for the Locomotor subscale, 0.84 to 0.997 for the Ball Skills subscale, and 0.92 to 0.996 for the Total TGMD-3 score. Pearson correlation coefficients were from 0.92 to 0.93 for the Locomotor subscale, 0.81 to 0.94 for the Ball Skills subscale, and 0.90 to 0.95 for the Total TGMD-3. Overall, the test–retest reliability was rated as sufficient, with high-quality evidence (Table 2).

Nine studies evaluated the inter-rater reliability of the TGMD-3 ([7]; [17]; [26]; [30]; [31]; [34]; [53]; [55]). The ICCs ranged from 0.82 to 0.97 for the Locomotor subscale, 0.778 to 0.98 for the Ball Skills subscale, and 0.842 to 0.98 for the total TGMD-3. These results were deemed sufficient, and the quality of evidence was judged to be high, as all studies were identified as having very good methodological quality (Table 2).

Intra-rater reliability was assessed in nine studies ([3]; [9]; [17]; [26]; [30]; [31]; [34]; [53]; [55]). The ICCs ranged from 0.865 to 0.988 for the Locomotor subscale, 0.85 to 0.99 for the Ball Skills subscale, and 0.90 to 0.99 for the Total TGMD-3. As all ICC values exceeded 0.7, the overall rating was deemed sufficient, with high-quality evidence (Table 2). Collectively, the findings indicate that the TGMD-3 exhibits sufficient reliability across test–retest, inter-rater, and intra-rater measures.

#### 3.3.5. Measurement Invariance

Six studies assessed the measurement invariance of the TGMD-3 across different groups (gender, age, and disability) using multi-group confirmatory factor analysis (MCFA) and differential item functioning (DIF) ([27]; [28]; [31]; [43]; [51]; [55]) (Table 2).

Four studies found no significant differences across gender groups using MCFA, and one study using DIF also found no significant differences, suggesting sufficient measurement invariance across genders. The quality of evidence for these studies was judged to be high.

For age group comparisons, one study found no significant differences using MCFA, and another using DIF also reported no significant differences, indicating sufficient measurement invariance across age groups. Both studies were rated as high-quality evidence. Additionally, one study examined measurement invariance between children with and without disabilities, finding no significant differences via MCFA. The quality of evidence for this study was also rated as high. Taken together, the findings suggest that the TGMD-3 has sufficient measurement invariance across gender, age, and disability groups, supported by high-quality evidence.

#### 3.3.6. Hypothesis Testing for Construction Validity

Two studies evaluated the construct validity of the TGMD-3 ([7]; [55]). These studies assessed the construct validity by correlating the TGMD-3 with similar domain instruments such as Test of Gross Motor Development-2 (TGMD-2), Movement Assessment Battery for Children-2 (M-ABC2), and German Youth Games ball-throwing distance performance (GYGBT).

One study assessed the convergent validity of the TGMD-3 with the TGMD-2 ([7]), reporting Pearson correlation coefficients of 0.98 for the Locomotor subtest, 0.98 for the Ball Skills subtest, and 0.99 for the Total scale, indicating sufficient convergent validity with high-quality evidence.

Another study assessed the divergent validity of the TGMD-3 with the M-ABC2 and the concurrent validity of the TGMD-3 with the GYGBT ([55]). The overall rating for these tests was insufficient, with correlation coefficients below 0.5. Despite this, the quality of evidence was still rated as high. Overall, the construct validity of the TGMD-3 was deemed inconsistent, with moderate-quality evidence due to divergent findings (Table 2).

#### 3.3.7. Responsiveness

Two studies examined the responsiveness of the TGMD-3 ([39]; [44]). One study compared the FMS performance of typically developing children with that of children with Attention Deficit Hyperactivity Disorder (ADHD), Autism Spectrum Disorder (ASD), Intellectual Disability (ID), and Language or Articulation Disorders (LAD) ([39]). The TGMD-3 demonstrated sufficient responsiveness in comparisons between typically developing children and those with ADHD, ASD, and ID (*p* < 0.05). However, responsiveness was insufficient in comparisons between typically developing children and those with LAD. The quality of evidence for these results was rated as low due to significant bias. The second study assessed responsiveness by comparing FMSs before and after an intervention ([44]), showing sufficient results. However, the quality of evidence remained low due to bias concerns (Table 2).

## 4. Discussion

To the best of our knowledge, this is the first systematic review to employ the COSMIN methodology to assess the measurement properties of the Test of Gross Motor Development-3 (TGMD-3). This review synthesised data from 23 studies, evaluating the TGMD-3 across various psychometric properties.

In accordance with the COSMIN guidelines, content validity holds the utmost importance among all measurement properties for any instrument or scale ([35]). [8] ([8]) highlight three primary sources for establishing content validity: the existing literature, judgement from representatives of the target population, and expert evaluation ([8]). Although expert judgement is commonly used ([4]), the COSMIN methodology emphasises the integration of both patient (or population) and expert viewpoints to assess three crucial aspects of content validity: relevance (the pertinence of all items to the construct within a specific context), comprehensiveness (the inclusion of all essential aspects of the construct), and comprehensibility (the understanding of items as intended by the population) ([47]).

In this review, four studies focused on evaluating the content validity of the TGMD-3. ([30]; [31]; [34]; [53]). These studies primarily evaluated the relevance and comprehensibility of the TGMD-3 through expert consultations. However, no study incorporated feedback from the target population (children) due to the nature of the TGMD-3, where children are required only to follow evaluator instructions to complete the movement tasks, rendering direct comprehension of the items less critical ([34]; [53]). Therefore, although the content validity assessment did not involve participants, the relevance and comprehensibility of the TGMD-3 were still deemed sufficient.

The aspect of comprehensiveness, which ensures that no key facets of the construct are missing, was notably absent from the included studies ([47]). Since no studies have specifically evaluated this component, a thorough assessment of the overall content validity of the TGMD-3 cannot be provided. It is recommended that future research endeavour to fill this gap by investigating whether the TGMD-3 comprehensively encompasses all crucial aspects of gross motor development across varied populations and settings.

Structural validity pertains to how well the scores of a scale capture the dimensional characteristics of the underlying construct it is intended to measure ([35]). In the case of TGMD-3, there is debate regarding its dimensionality. While some studies support a bifactor structure, others advocate for a one-factor model. Our systematic review, utilising COSMIN’s approach to synthesising evidence, found sufficient moderate-quality evidence favouring a bifactor structure, with inconsistent moderate-quality evidence supporting a one-factor model. Thus, the bifactor structure appears to be a more appropriate representation of the TGMD-3’s dimensionality. This finding aligns with previous research, such as that by [19] ([19]), who also concluded that the bifactor model provided a better fit compared to both one-factor and two-factor models ([19]).

Cronbach’s alpha values exceeding 0.7 are typically regarded as indicative of adequate internal consistency ([46]). In the studies included in this review, 92% of them demonstrated that the internal consistency of TGMD-3 is sufficient. According to the COSMIN manual, if at least 75% of the results meet the threshold for sufficiency, the measurement property can be rated as sufficient. However, the quality of evidence must be downgraded due to inconsistent findings, reducing it from high-quality evidence to moderate-quality evidence ([35]). Therefore, based on our findings, moderate-quality evidence supports the conclusion that the internal consistency of TGMD-3 is sufficient. This outcome, as per COSMIN guidelines, indicates that the inter-relatedness among the items on the TGMD-3 scale is adequately high ([35]).

According to the COSMIN guidelines, reliability refers to the degree to which scores remain consistent across various conditions, such as different time intervals (test–retest reliability), different raters at the same time (inter-rater reliability), and the same rater at different times (intra-rater reliability) ([35]). Our results reinforce that there is high-quality evidence supporting the sufficiency of reliability (test–retest, intra-rater, and inter-rater reliability) of the TGMD-3. These findings are consistent with another systematic review by [41] ([41]), which also found the TGMD-3 to be sufficiently reliable ([41]).

As outlined in the COSMIN manual, cross-cultural validity refers to “the extent to which the performance of items on translated or culturally adapted measurement instruments accurately reflects the performance of the items in the original version of those instruments” ([35]). Our analysis revealed that no studies have evaluated the cross-cultural validity of the TGMD-3 using the COSMIN-recommended methodology. Consequently, there is uncertainty regarding the ability of translated versions of the TGMD-3 to accurately represent the original instrument. Therefore, we recommend validating the cross-cultural validity of these translated versions before utilising them to assess children’s fundamental motor skills (FMSs).

Measurement invariance refers to the consistency of a test’s scores across different groups (e.g., gender, age) ([54]). Our assessment demonstrated that high-quality evidence supports the sufficiency of TGMD-3’s measurement invariance across three groups: different genders, ages, and children with and without disabilities. These results suggest that TGMD-3 is appropriate for use across diverse populations, ensuring it is a robust tool in various demographic settings.

Responsiveness refers to a scale’s ability to detect meaningful changes in the construct being measured over time ([35]). Two studies included in our review ([39]; [44]) evaluated the responsiveness of TGMD-3 by comparing the fundamental movement skill (FMS) scores of different child groups (e.g., children with and without ADHD, ASD, ID, and those before and after a physical education intervention). While the TGMD-3 showed sufficient responsiveness in most comparisons, it was found to be insufficient in the comparison between children with and without LAD. The overall quality of evidence for these studies was judged to be low due to severe biases. Notably, these bias due to that the studies employed a paired t-test, which is not recommended by COSMIN as it measures statistical significance rather than valid change. This method is considered inappropriate for assessing responsiveness ([35]).

Based on COSMIN guidelines, a measurement instrument or scale can be categorised as “A” if it has sufficient content validity (regardless of evidence quality level) or sufficient internal consistency (at least low-quality evidence). Conversely, instruments with high-quality evidence demonstrating inadequate measurement properties are classified as “C”. Instruments that do not meet the criteria for either “A” or “C” are classified as “B” ([40]). Our review found that while the internal consistency of TGMD-3 was sufficient with moderate-quality evidence, the lack of evaluation regarding the comprehensiveness of the TGMD-3’s content validity prevents a full assessment. Therefore, the TGMD-3 can be categorised as “B”. Instruments in this category hold promise for recommendation but require further research to validate their quality comprehensively. We recommend that future studies specifically address the comprehensiveness of items of the TGMD-3 to strengthen its overall utility and validity.

This study has several limitations. Firstly, our screening was limited to studies published in English and Chinese, potentially leading to the omission of important research on the cross-cultural validity of TGMD-3. Cross-cultural validity, defined as “the degree to which the performance of the items on a translated scale reflects the performance of the items in the original version”, is especially relevant when translating the TGMD-3 into different languages. The outcomes of cross-cultural validation studies are often published in the local language of the study; thus, excluding non-English- and non-Chinese-language studies may have restricted the comprehensiveness of our review. To thoroughly assess cross-cultural validity, future reviews should consider including studies published in a broader range of languages.

Secondly, our study focused solely on evaluating the measurement properties of the original version of the TGMD-3, excluding other versions such as the TGMD-3 short form ([14]) and the Motor Skills Sequential Pictures (MSSP) version of the TGMD-3 ([12]). As different versions of the TGMD-3 emerge, each with potentially unique measurement properties, they warrant separate examination to determine their reliability and validity in different contexts and populations.

Drawing from our findings, we propose the following recommendations for future research endeavours. Initially, in accordance with the COSMIN manual, a measurement instrument or scale can be categorised as “A” if it has sufficient content validity (regardless of evidence quality level) or at sufficient internal consistency (at least low-quality evidence) ([40]). Our review suggests that, with additional research verifying the comprehensiveness of the TGMD-3’s items, the tool could potentially be upgraded to an “A” classification. Instruments in this category provide robust, trustworthy results. Therefore, we suggest that future studies should specifically examine the comprehensiveness of the TGMD-3’s content validity to provide more conclusive evidence on this aspect. Secondly, further investigations into the TGMD-3’s responsiveness and measurement error should be conducted using the COSMIN methodology. These properties are essential for assessing the ability of the TGMD-3 to detect meaningful changes over time and ensuring the accuracy of measurements. By expanding the evidence base on these measurement properties, future research can solidify the TGMD-3’s utility across diverse settings and populations, ultimately enhancing its applicability and reliability in both clinical and educational environments.

## 5. Conclusions

The assessment of the TGMD-3 using the COSMIN methodology demonstrated that its test–retest reliability, inter-rater reliability, intra-rater reliability, and measurement invariance were sufficient, supported by high-quality evidence. The content validity, in terms of relevance and comprehensibility, was also deemed sufficient with high-quality evidence. Furthermore, a bifactor structure emerged as a more suitable model for TGMD-3 compared to a one-factor structure, with structural validity and internal consistency supported by moderate-quality evidence. However, the results of hypothesis testing for construct validity were inconsistent, and responsiveness was found to be inconsistent, supported by low-quality evidence.

Overall, TGMD-3 was classified as a “B” grade instrument. Instruments in this category show potential for recommendation but require further research to comprehensively evaluate their measurement properties. In particular, future studies should aim to assess the comprehensiveness of items of the TGMD-3 to strengthen its overall validity and reliability across diverse populations and contexts.

## Figures and Tables

**Figure 1 behavsci-15-00062-f001:**
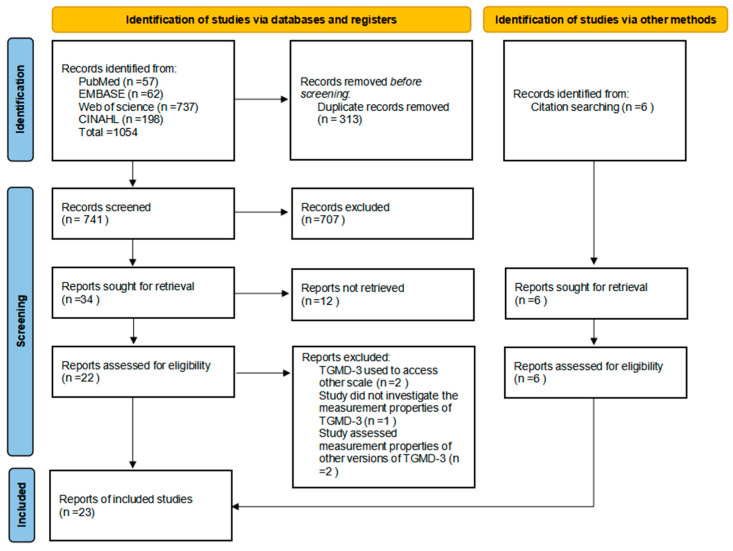
Flow diagram of article selection according to PRISMA.

**Table 1 behavsci-15-00062-t001:** Basic characteristics of the included articles.

Author (Year)	Population Characteristics				Research Characteristics of TGMD-3	
	N	Age (Years)	Sex (M/F)	Studied Population	Country/ Region	Measurement Properties
[3] ([3])	35	4–10	22/13	Typically developing children and children with ASD	Australia	IC, TR, IER, IAR, SV
[7] ([7])	66	12.93 ± 2.4	41/25	Children with visual impairment	USA	IC, TR, CoV, SV
[5] ([5])	302	13 ± 2.5	175/127	Children with visual impairment	USA	SV
[9] ([9])	25	9.16 ± 1.31	60% girls	Typically developing children	Spain	IAR, IER
([14])	1608	9.2 ± 2.04	47% girls	Typically developing children	Ireland	SV
[17] ([17])	178	3–11	47.5% girls	Typically developing children	Spain	IC, SV, IER, IAR
[19] ([19])	862	6.5 ± 2.23	49% girls	Typically developing children	USA	SV, IC
[26] ([26])	10	6.57 ± 2.51	6/4	Typically developing children	USA	IAR, IER
[28] ([28])	1075	3–11	565/510	Children with mental disorders and typically developing children	Italy	MI, IC
[27] ([27])	5210	8.38 ± 1.97	48% girls	Typically developing children	Italy	MI, SV, TR
[29] ([29])	127	5–11	70/57	Typically developing children	Canada	SV
[30] ([30])	348	6–10	48.6% girls	Typically developing children	Peru	SV, IC, SV, IC, TR
[31] ([31])	452	7.32 ± 2.16	50.4% girls	Typically developing children	Slovenia	CV, IAR, IER, TR, SV, IC, MI
[32] ([32])	146	6.8 ± 2.23	53.4% girls	Typically developing children	Bosnia and Herzegovina	IC, SV
[34] ([34])	1600	6.56 ± 2.29	50% girls	Typically developing children	Iran	CV, IC, IR, IER, TR, SV
[39] ([39])	170	3–11	122/48	Disabled children and typically developing children	USA	Re
[42] ([42])	290	11–13	180/110	Typically developing children	Indonesia	CV, IC
[43] ([43]	496	7.23 ± 2.03	53.8% girls	Typically developing children	Iran	SV, MI
[44] ([44])	48	5.10 ± 0.74	28/20	Children with Down syndrome	USA	Re
[51] ([51])	989	3–10.9	498/491	Typically developing children	Brazil	MI
[53] ([53])	597	3–10	295/320	Typically developing children	Brazil	SV, CV, IER, IAR, TR, IC
[55] ([55])	189	7.15 ± 2.02	99/90	Typically developing children	Germany	IER, IAR, TR, IC, MI, CoV, SV
[57] ([57])	807	6.33 ± 2.09	47.5% girls	Typically developing children	USA	IC, TR, SV

Note: M/F = male/female; CV: content validity; SV: structural validity; IC: internal consistency; MI: measurement invariance; TR: test–retest reliability; IeR: inter-rater reliability; IaR: intra-rater reliability; Re: responsiveness; CoV: concurrent validity.

**Table 2 behavsci-15-00062-t002:** Summary of the findings.

Measurement Property	Summary or Pooled Results	Overall Rating	Quality of Evidence
**Content validity**	**Content validity** ([30]; [31]; [34]; [53])		
	Relevant ([30]; [31]; [34]; [53])	**Sufficient (+)**	**High**: multiple very good studies
	Comprehensible ([30]; [31]; [34]; [53])	**Sufficient (+)**	**High**: multiple very good studies
**Structural validity**	**Structural validity** ([7]; [5]; [17]; [19]; [27]; [29]; [30]; [31]; [32]; [34]; [43]; [53]; [57])		
	Bifactor structure ([7]; [5]; [17]; [19]; [27]; [29]; [30]; [31]; [32]; [34]; [43]; [53]; [57])	**Qualitative summary: Sufficient (+)** 79% supported	Moderate: multiple very good studies, inconsistent results
	One-factor structure ([17]; [19]; [31]; [34]; [57])	**Qualitative summary:** inconsistent (±)	Moderate: multiple very good studies, inconsistent results
**Internal consistency**	**Internal consistency** ([3]; [7]; [17]; [19]; [28]; [30]; [31]; [32]; [34]; [42]; [53]; [55]; [57])	**Qualitative summary: Sufficient (+)**	Moderate: multiple very good studies, inconsistent results
	Locomotor Subtest α: 0.63–0.92	**Sufficient (+)** 92% supported	
	Ball skills Subtest α: 0.60–0.95	**Sufficient (+)** 92% supported	
	Total TGMD-3 α: 0.74–0.96	**Sufficient (+)**	
**Reliability**	**Reliability** ([3]; [7]; [9]; [17]; [26]; [27]; [30]; [31]; [34]; [53]; [55]; [57])		
	**Test–retest reliability** ([27]; [30]; [31]; [34]; [53]; [55]; [57])	**Qualitative summary: Sufficient (+)**	**High: all studies are very good**
	Locomotor Subtest		
	ICC: 0.81–0.996 ([3]; [27]; [30]; [31]; [55]; [57])	**Sufficient (+)**	
	r: 0.92–0.93 ([34]; [53])	**Sufficient (+)**	
	Ball Skills Subtest		
	ICC: 0.84–0.997 ([3]; [27]; [30]; [31]; [55]; [57])	**Sufficient (+)**	
	r: 0.81–0.94 ([34]; [53])	**Sufficient (+)**	
	Total TGMD-3		
	ICC: 0.92–0.996 ([3]; [27]; [30]; [31]; [55]; [57])	**Sufficient (+)**	
	r: 0.90–0.95 ([34]; [53])	**Sufficient (+)**	
	**Inter-rater reliability** ([7]; [17]; [26]; [30]; [31]; [34]; [53]; [55])	**Qualitative summary: Sufficient (+)**	**High: all studies are very good**
	Locomotor Subtest		
	ICC: 0.82–0.97	**Sufficient (+)**	
	Ball Skills Subtest		
	ICC: 0.778–0.98	**Sufficient (+)**	
	Total TGMD-3		
	ICC: 0.842–0.98	**Sufficient (+)**	
	**Intra-rater reliability**: ([3]; [9]; [17]; [26]; [30]; [31]; [34]; [53]; [55])	**Qualitative summary: Sufficient (+)**	**High: all studies are very good**
	Locomotor Subtest		
	ICC: 0.865–0.988	**Sufficient (+)**	
	Ball skills Subtest		
	ICC: 0.85–0.99	**Sufficient (+)**	
	Total TGMD-3		
	ICC: 0.90–0.99	**Sufficient (+)**	
**M** **easurement invariance**	**Measurement invariance** ([27]; [28]; [31]; [43]; [51]; [55])	**Qualitative summary: Sufficient (+)**	**High: all studies are very good**
	Across gender groups: No important differences ([27]; [31]; [43]; [55]) OR no important DIF ([51])	**Sufficient (+)**	
	Across age groups: No important differences ([27]) OR no important DIF ([51])	**Sufficient (+)**	
	Across with and without disability groups: No important differences ([28])	**Sufficient (+)**	
**Hypothesis testing for construct validity**	**Hypothesis testing for construct validity** ([7]; [55])	**Qualitative summary:** inconsistent (±)	Moderate: multiple very good studies, inconsistent results
	TGMD-3 and TGMD-2 ([7]) Convergent validity: Locomotor Subtest r: 0.98 Ball Skills Subtest r: 0.98 Total Scale r: 0.99	**Sufficient (+)**	
	TGMD-3 and M-ABC2 ([55]) Divergent Validity: Locomotor Subtest r: 0.22–0.33 Ball Skills Subtest r: 0.23–0.30	**insufficient (−)**	
	TGMD-3 and GYGBT ([55]) Concurrent validity r: 0.36	**insufficient (−)**	
**Responsiveness**	Responsiveness ([39]; [44])	**Qualitative summary:** inconsistent (±)	**Low: severe bias**
	Hypothesis testing: comparison between subgroups ([39])		
	Children with and without ADHD: *p* < 0.05	**Sufficient (+)**	**Low: severe bias**
	Children with and without LAD: *p* > 0.534	**Insufficient (−)**	**Low: severe bias**
	Children with and without ASD: *p* < 0.001	**Sufficient (+)**	**Low: severe bias**
	Children with and without ID: *p* < 0.001	**Sufficient (+)**	**Low: severe bias**
	hypotheses testing: before and after intervention ([44]) Locomotor Subtest: (*p* < 0.01) Ball Skills Subtest: (*p* < 0.01)	**Sufficient (+)**	**Low: severe bias**

Note: TGMD-2 = Test of Gross Motor Development-2; M-ABC2 = Movement Assessment Battery for Children-2; GYGBT = German Youth Games ball-throwing distance performance; ADHD = Attention Deficit Hyperactivity Disorder; LAD = Language or Articulation Disorders; ASD = Autism Spectrum Disorder; ID = Intellectual Disability.

## Data Availability

All the data that support the findings of this study are available from the corresponding author upon reasonable request.

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
