# Peer review of "Assessing the Measurement Properties of the Test of Gross Motor Development-3 Using the COSMIN Methodology—A Systematic Review"

_behavsci, 2025, doi:10.3390/bs15010062_

Round 1

Reviewer 1 Report

Comments and Suggestions for Authors

This is a great systematic review study of the psychometric properties of the Test of Gross Motor Development-3. The manuscript includes impressive and robust information on the validity and reliability of the TGMD-3 across a range of cultural contexts and children. The authors have followed standard procedures such as the COSMIN checklist and have presented the PRISMA flowchart, as well. The GRADE procedure has also been applied. The main manuscript is supplemented with a helpful file explaining the key definitions of the facets of validity and reliability considered here. Please find below some minor feedback for improving the quality of the paper.

·        Kindly distinguish between fine and gross motor development in the literature review.

·        It would be really helpful to add a short section describing the key developmental milestones of the gross motor development in the children’s age group under study (ages 3-11).

·        I would appreciate some more context on the initial development of the TGMD-3 scale, particularly information related to the initial country of construction, the language of initial validation, and initial evidence on its psychometric properties.

·        The information from line 63 could be a different subsection inside the literature review.

·        Lines 68-75, please explain more what this conflicting evidence is about. How are the two factors named (locomotor vs ball skills)? Also, consider explaining a bit more about the conflict in terms of the reliability of the measure.

·        Please add inside the method’s section the language(s) used in the searching engine.

·        Instead of ‘normal’ children, consider the term ‘typically developing’ children, as more developmentally appropriate.

·        Section 3.3.5: the authors mention ‘cross-cultural’ validity; however, ‘cross-cultural’ validity refers to a measure functioning (i.e., being understood and scored) in the same way across different cultural groups (e.g., countries, ethnic groups). Therefore, the use of ‘cross-cultural’ in that particular section is inappropriate.

·        Please also explain what a bifactor structure means since some readers might be unfamiliar with this structural specification.

·        I think the discussion needs to reflect the fact that there is not any evidence, as far as I can ascertain, about the cross-country/cultural comparability of the TGMD-3. This means that researchers apply an US-specific model of gross motor development to other contexts, where children’s gross motor development might not follow exactly the same milestones as in the US.

I hope the authors will find my comments helpful!

Author Response

Kindly distinguish between fine and gross motor development in the literature review.

Response: Thank you for your suggestion. This recommendation is highly useful for providing a clearer description of the research background. Now, we have added descriptions related to gross motor skills and fine motor skills in the section L40-45 and L48-49.

It would be really helpful to add a short section describing the key developmental milestones of the gross motor development in the children’s age group under study (ages 3-11).

Response: We are thankful to the Reviewer for this interesting and useful comment. We have incorporated information on key motor development milestones for young children in the section L49-56. The addition of these details makes it easier for readers to understand the significant role of FMS development in the growth process of young children and further emphasizes the importance of screening FMS development in this age group.

I would appreciate some more context on the initial development of the TGMD-3 scale, particularly information related to the initial country of construction, the language of initial validation, and initial evidence on its psychometric properties.

Response: Thank you for your suggestion, Now, we have added more detailed information in L73-80 to the original version of the TGMD, including the initial country of construction, the language of initial validation, and initial evidence on its psychometric properties. The inclusion of these details enhances readers' understanding of the three versions of the TGMD as measurement tools.

The information from line 63 could be a different subsection inside the literature review.

Response: We thank Reviewer for this comment. We agree with the reviewer's perspective. After careful consideration, we have decided to place this content in the next section (L91-92). Since excellent measurement properties are a prerequisite for the use of a scale, highlighting the widespread use of the TGMD-3 is more conducive to introducing the existing discussions on the measurement properties of TGMD-3 in the research context.

Lines 68-75, please explain more what this conflicting evidence is about. How are the two factors named (locomotor vs ball skills)? Also, consider explaining a bit more about the conflict in terms of the reliability of the measure.

Response: Thank you for your suggestion.This recommendation is highly useful. We apologize for our erroneous use of the term "conflicting," which may have led to misunderstanding. "Inconsistent" would be a more appropriate term, and we have replaced "conflicting" with "inconsistent" at L100 and L110. Additionally, we have included information related to the naming of the two factors at L101-103. All the studies we included demonstrated adequate reliability of the TGMD-3, so we could not find conflicting information in the evidence regarding the reliability of the TGMD-3.

Please add inside the method’s section the language(s) used in the searching engine.

Response: We thank Reviewer for this suggestion. We have added information on the language types used during the search process across multiple databases for the measurement properties of the TGMD-3 at L136.

Instead of ‘normal’ children, consider the term ‘typically developing’ children, as more developmentally appropriate.

Response: We thank the reviewer for this comment. We have replaced "normal children" with "typically developing children" in Table 1 and L164.We agree with the reviewer's perspective. Using the term "typically developing" children is a more precise description compared to "normal" children.

Section 3.3.5: the authors mention ‘cross-cultural’ validity; however, ‘cross-cultural’ validity refers to a measure functioning (i.e., being understood and scored) in the same way across different cultural groups (e.g., countries, ethnic groups). Therefore, the use of ‘cross-cultural’ in that particular section is inappropriate.

Response: Thank you for your suggestion. We have removed the term "cross-cultural validity" from the 3.3.5 section. According to the COSMIN manual, it is recommended to evaluate "cross-cultural validity" together with "measurement invariance." Since the studies we included did not involve research on "cross-cultural validity," it is reasonable to omit “cross-cultural validity” the 3.3.5 section.

Please also explain what a bifactor structure means since some readers might be unfamiliar with this structural specification.

Thank you for this very useful suggestion. We have incorporated an explanation of the "bifactor structure" at L101-102 to facilitate a more accessible understanding of this term for our readers.

I think the discussion needs to reflect the fact that there is not any evidence, as far as I can ascertain, about the cross-country/cultural comparability of the TGMD-3. This means that researchers apply an US-specific model of gross motor development to other contexts, where children’s gross motor development might not follow exactly the same milestones as in the US.

Response: We are thankful to the Reviewer for this interesting comment. We have incorporated discussions related to cross-cultural validity at L472-480. We have elaborated on the current situation where various translated versions of the TGMD-3 lack validation for cross-cultural validity and provided corresponding suggestions for researchers who intend to use these translated versions of the TGMD-3 in the future.

Reviewer 2 Report

Comments and Suggestions for Authors

The manuscript titled “Systematic Review of Measurement Properties of the Test of Gross Motor Development-3 Using COSMIN Methodology” authored by Yuanye Zhu, Jing Wang, Yaru Ding, Yongdong Qian, Mallikarjuna Korivi, Qian Chen, and Weibing Ye, which reflects their dedicated research and scholarly efforts fulfills the criteria for publication in Behavioral Sciences and should thus be accepted after minor revisions. The review aims to consolidate existing research on the TGMD-3 and evaluate the quality of evidence using the COSMIN methodology, providing valuable insights for researchers and practitioners in physical education and child development. By critically analyzing the reliability, validity, and overall measurement properties of the TGMD-3, the review seeks to assess the instrument's trustworthiness in accurately measuring fundamental movement skills in children. However, some minor technical improvements are welcome to enhance the presentation of the findings, which will further strengthen the overall impact of the work.

1.      Line 122 - Consider putting a direct link to your study protocol registered in PROSPERO (https://www.crd.york.ac.uk/prospero/display_record.php?ID=CRD42024600851).

2.      Please ensure consistent spacing throughout the manuscript, particularly around symbols such as "±" (e.g., "10 ± 2" instead of "10±2"), around equal signs (e.g., "x = 5" instead of "x=5"), p values (e.g., "p < 0.05" instead of " p< 0.05") and between numbers and percentages (e.g., "20 %" instead of "20%") in text as well as tables.

3.      Please add DOI for all references where possible. For example, reference number nine (http://dx.doi.org/10.46827/ejes.v8i10.3928).

Author Response

Comments1: Line 122 - Consider putting a direct link to your study protocol registered in PROSPERO (https://www.crd.york.ac.uk/prospero/display_record.php?ID=CRD42024600851).

Response1: Thank you for your suggestion. We have added a direct link to our study protocol registered in PROSPERO as your suggestion (L160). This will facilitate readers in easily accessing the relevant information.

Comments 2: Please ensure consistent spacing throughout the manuscript, particularly around symbols such as "±" (e.g., "10 ± 2" instead of "10±2"), around equal signs (e.g., "x = 5" instead of "x=5"), p values (e.g., "p < 0.05" instead of " p< 0.05") and between numbers and percentages (e.g., "20 %" instead of "20%") in text as well as tables.

Response2: We are apologizing for the erroneous. We have now corrected the errors you mentioned and conducted a thorough review of the manuscript to avoid similar issues in the future.

Comments 2: Please add DOI for all references where possible. For example, reference number nine (http://dx.doi.org/10.46827/ejes.v8i10.3928).

Response3: Thank you for your suggestion. We re-searched each cited reference and added the DOIs we could find to the bibliography section.
